# Different Methods and Formulations of Drugs and Vaccines for Nasal Administration

**DOI:** 10.3390/pharmaceutics14051073

**Published:** 2022-05-17

**Authors:** Junhu Tai, Munsoo Han, Dabin Lee, Il-Ho Park, Sang Hag Lee, Tae Hoon Kim

**Affiliations:** Department of Otorhinolaryngology-Head & Neck Surgery, College of Medicine, Korea University, Seoul 02841, Korea; junhu69@korea.ac.kr (J.T.); mshan35@gmail.com (M.H.); dabin425@korea.ac.kr (D.L.); parkil5@korea.ac.kr (I.-H.P.); sanghag@kumc.or.kr (S.H.L.)

**Keywords:** drug delivery, intranasal, vaccine, immunity, nanotechnology

## Abstract

Nasal drug delivery is advantageous when compared with other routes of drug delivery as it avoids the hepatic first-pass effect, blood–brain barrier penetration, and compliance issues with parenteral administration. However, nasal administration also has some limitations, such as its low bioavailability due to metabolism on the mucosal surface, and irreversible damage to the nasal mucosa due to the ingredients added into the formula. Moreover, the method of nasal administration is not applicable to all drugs. The current review presents the nasal anatomy and mucosal environment for the nasal delivery of vaccines and drugs, as well as presents various methods for enhancing nasal absorption, and different drug carriers and delivery devices to improve nasal drug delivery. It also presents future prospects on the nasal drug delivery of vaccines and drugs.

## 1. Introduction

The nose regulates inhaled air through filtration and humidification and protects the airway from potentially harmful particles. It also acts as the sensory organ that is responsible for smell [1]. The nose is a very valuable route of administration, and the high vascularization and high permeability of the nasal mucosa also make it possible to administer drugs through this route [2]. Nasal administration has many advantages when compared with oral administration, such as a rapid onset of action, less drug degradation, and a high rate of absorption. When compared with intravenous administration, it has high patient compliance, self-administration, and direct nose-to-brain delivery by bypassing the blood–brain barrier via the olfactory nerve pathways [3,4].

Nasal administration, with its many advantages, naturally has made researchers interested in the nasal route of drug delivery. However, mucociliary clearance poses significant obstacles to the systemic delivery of drugs via the nasal cavity [5]. The entire surface of the nasal cavity is covered by a mucus layer [6]. Mucociliary clearance rapidly removes drugs from the absorption site. In addition to mucociliary clearance, hair in the nostrils, sneezing, and coughing also greatly reduce the number of particles available to enter the human body through the mucosal surface [7]. The use of mucosal adhesives such as chitosan overcomes some obstacles to mucociliary clearance. It can increase the efficacy of drugs and the immunogenicity of vaccines by prolonging the residence time of drugs or antigens at immune effector sites [8,9]. Researchers have developed a nasal vaccine for influenza, which has the mucosal adhesion characteristics of chitosan [10]. In addition to avoiding the capture and clearance of mucosal cilia, it is also important to protect drugs from enzymatic degradation [11]. Using liposomes or micelles to form protective shells, or adding enzyme inhibitors are feasible methods to increase retention time in the nose, and subsequently enhance absorption [12]. In addition to the above disadvantages, as shown in Table 1, nasal drug delivery also has the disadvantages such as irreversible damage to the nasal mucosa caused by the ingredients added in the formula, which is not applicable to all drugs, and is affected by nasal conditions such as allergy conditions [13,14].

After years of development, considerable progress has been made in nasal drug delivery devices. The nasal spray is the most widely used device. It has the advantages of relative simplicity and low manufacturing cost. However, it also has shortcomings, such as an inaccurate dose and an insufficient depth of administration. Acoustic wave nebulization and other biological materials can compensate for these shortcomings. Research on new drug delivery systems such as nanoparticles and nanofibers for the delivery and controlled release of poorly permeable molecules is also in progress [15]. This review comprehensively describes the mucosal environment for nasal administration, explains and classifies the drugs entering the systemic circulation via the nasal route of delivery and vaccines that can act by activating systemic and local mucosal immune responses, and introduces newly developed and developing materials for nasal drug carriers. The advantages and disadvantages of nasal drug delivery methods were analyzed, which provides some information and tips for the study of new nasal administration methods.

### 1.1. Mucosal Environment of Nasal Cavity

The nasal mucosa consists of the epithelium, basement membrane, and lamina propria. There are four main cell types in nasal mucosa: basal cells, goblet cells, ciliated columnar cells, and non-ciliated columnar cells [16]. Basal cells are found only on the basement membrane, and the other three cell types are found on the whole apical epithelial surface (Figure 1). The full absorption of drug active ingredients needs to control for their release curve when they pass through multiple biological barriers. These barriers include: the mucus layer, epithelial layer, stroma and basement membrane, and capillary endothelium [17]. The first barrier is the mucus layer. The drug needs to dissolve or pass through the mucus layer quickly, because the cilia will remove the drug from the absorption site. The second barrier is the epithelial cell membrane. Most drugs are mainly absorbed through cross diffusion and penetrate through the epithelial cell membrane. If drug molecules pass through the stroma and basement membrane, the fourth barrier they face will be the capillary endothelium. When compared with locally acting drugs, passing through this barrier is more important for systematically targeted drugs. The pharynx is composed of the nasopharynx, oropharynx, and hypopharynx [18]. Nasopharynx-associated lymphoid tissue (NALT) is composed of lymphoid tissue in the Waldeyer ring, including adenoids, lingual tonsils, and palatal tonsils. It is an attractive inductive site; intranasal vaccination can stimulate the immune response of NALT [19].

### 1.2. Pathways and Destinations of Nasally Administered Drugs

There are three routes for drugs to enter the brain via the nose (Figure 2) [20]. The first route for drugs to enter the brain is via the olfactory pathway. Drugs enter the brain through the olfactory epithelium, which is composed of basal cells, supporting cells, and olfactory nerve cells; they can be then transported to the olfactory bulb through the olfactory nerve [21]. The first route is divided into three pathways, including the intracellular transport pathway after the drug’s internalization into neurons, extracellular transport across intercellular spaces, and intercellular transport across basal epithelial cells. Drugs using the neuronal pathway are transported to the olfactory bulb through endocytosis or pinocytosis, are released, and then distributed to different brain regions through exocytosis [22]. In the extraneuronal pathway, substances administered intranasally first pass through the gap between olfactory neurons in the olfactory epithelium and are then transported to the olfactory bulb [23]. After reaching the olfactory bulb, substances may diffuse into other areas of the brain, which may also be promoted by “perivascular pumps”, driven by the arterial pulse [24]. Drugs transported through the extraneuronal pathway can reach the olfactory bulb and other brain regions in only a few minutes, which is much faster than the speed that takes hours to days within neurons, indicating that the main route of the nose to brain transport is extracellular transport [25]. The second route for drugs to enter the brain is via the trigeminal pathway. The trigeminal nerve, which innervates the olfactory epithelium and mucosa, constitutes an additional but less important pathway for direct drug transport from the nose to the brain [26], and cannot be accurately measured, because part of the trigeminal nerve enters the brain through the sieve plate, which is adjacent to the olfactory pathway [27]. Although the contribution of the trigeminal pathway to drug delivery from the nasal cavity to the brain seems to be less than that of the olfactory pathway, it is reported that insulin-like growth factor I is transported through the axon of the trigeminal nerve, which confirms the existence of the trigeminal pathway [28]. The third route is via a peripheral pathway. Drugs enter the systemic circulation via vascular absorption and subsequently enter the brain through the blood–brain barrier. This route has several limitations, including the excretion of drugs via the kidneys, the binding of drugs to plasma proteins, degradation of drugs by plasma proteases, and other potential peripheral effects [29]. For low-molecular-weight lipophilic molecules and drugs that easily permeate the blood–brain barrier, this process is very fast. However, for hydrophilic drugs with a high molecular weight and drugs that are rapidly eliminated from the blood flow, this pathway still has limitations, such as difficulty in passing through the blood–brain barrier [29,30]. Nasally administered drugs may also reach other destinations including the gastrointestinal tract, lungs, and lymphatic system [31]. Drugs may be swallowed into the oral cavity or gastrointestinal tract along the bottom of the nasal cavity and are absorbed by the gastrointestinal mucosa. Drugs also enter the lungs through inhalation and are absorbed into the systemic circulation. Similarly, many drugs are absorbed by the lymphatic system.

## 2. Drugs for Nasal Administration

Many drugs are administered through the nasal route, including corticosteroids, decongestants, antihistamines, and vaccines. In addition, various concentrations of saline are also used through the nasal route (Table 2). Among them, corticosteroids and antihistamines are the most commonly used and are the first-line drugs for the treatment of various types of rhinitis [32]. They are fast on-set of action, which is advantageous for locally active drugs. In addition, corticosteroids and antihistamines have lower systemic bioavailability when used locally; therefore, they have fewer adverse effects such as sedation, drowsiness, amnesia, and respiratory depression when compared with systemic use [33]. Additionally, they are potent at low doses, which is another advantage when compared with large doses of oral drugs [34]. The effective delivery of drugs to the central nervous system is the key to the treatment of brain tumors and neurodegenerative diseases such as stroke, Parkinson’s disease, and Alzheimer’s disease. Because intranasal delivery can enable drugs in passing through the blood–brain barrier, which is one of the most important barriers in the central nervous system, it can be considered as a development potential local delivery strategy [35].

### 2.1. Nasal Drugs

For a long time, corticosteroids have been widely used by clinicians for the local treatment of various rhinitis. After binding to the applicable receptor, they exert an anti-inflammatory effect through trans-activation or trans-inhibition [36]. Most corticosteroids in the market are second-generation. They are roughly divided into four categories: ciclesonide (Omnaris^®^), mometasone furoate (Nasonex^®^), fluticasone furoate (Avamys^®^), and fluticasone propionate (Flonase^®^). The active ingredients of these formulations are different; however, the excipients and mode of action are quite similar. In a four-week trial, once daily intranasal ciclesonide were found to be more effective than a placebo in improving nasal symptoms in patients with moderate and severe allergic rhinitis, and they were generally well tolerated. The intensity of most adverse events such as headache and epistaxis was mild to moderate [37]. In a 3-month randomized, double-blind study involving 550 allergic patients, patients were given mometasone furoate, fluticasone propionate, or a placebo once a day. Judging by the overall nasal symptoms evaluated by doctors, mometasone furoate and fluticasone propionate were more effective than the placebo, which was statistically significant. Their baseline reduction percentages were 37, 39, and 22%, respectively. There was no rapid drug resistance, and the treatment was well tolerated [38]. In another 6-week randomized controlled trial, 108 participants were assigned to fluticasone furoate twice a day for 6 weeks and found improvements in their nasal symptom scores and quality of life in rhinoconjunctivitis [39]. A comparison review of the efficacy of the second generation of nasal corticosteroids, such as ciclesonide, fluticasone furoate, fluticasone propionate, and mometasone furoate, indicates that there is not enough evidence that different types of corticosteroid sprays have different effects. There is no evidence that one type of intranasal steroid is more effective than another [40].

Antihistamines work by blocking the action of histamine, which causes many allergic symptoms. Azelastine is a widely used locally acting antihistamine undergoing extensive research [41]. Astelin^®^ and Astepro^®^ are azelastine products with high H1-receptor selectivity, and are currently available in the market. These products have slight differences in their excipient composition. The initial double-blind controlled trial of Astelin^®^ involved more than 200 patients with allergic rhinitis, who were randomly divided into an azelastine group or placebo group for 2 weeks. When compared with the placebo group, the symptoms of runny nose, itching nose, sneezing, blowing nose, and tears in the azelastine group were improved by more than 30%, which was statistically significant, and there were no safety problems [42]. Astepro^®^ is a newly formulated antihistamine; it was confirmed to be safe and well-tolerated in a one-year randomized study involving more than 800 patients with allergic rhinitis, showing little difference in efficacy from Astelin^®^. Headache, nasopharyngitis, and epistaxis were the most reported side effects. There was no evidence of increased nasal irritation when compared to the Astelin^®^, and there were no reports of perforation of the nasal septum, severe epistaxis, or ulcer [43]. Although corticosteroids are considered superior to antihistamines in terms of their therapeutic efficacy and safety, a previous study noted that they did not differ significantly in these areas [44]. A preliminary comparative study of azolastine nasal spray and fluticasone nasal spray found that the efficacy of azelastine was better than fluticasone and placebo in the early stage of administration, but there was no difference after 7 days of administration [45]. At the same time, some researchers also conducted a study on whether the combination of intranasal antihistamines and intranasal corticosteroids has better efficacy. It was found that the improvement of nasal symptoms was 27.1% in the fluticasone alone group, 24.8% in the azelastine alone group and 37.9% in the combination group, which was statistically significant, and could continuously improve symptoms with good tolerance [46].

Saline irrigation is an adjuvant therapy, which plays an important role in the treatment of chronic rhinosinusitis (CRS). Saline irrigation can dilute the mucus, reduce edema, and improve mucociliary clearance [47]. The commonly used saline solutions are divided into isotonic saline, hypotonic saline, and hypertonic saline, while Ringer’s lactate solution may also be used. The concentrations of sodium chloride in isotonic, hypotonic, and hypertonic saline are 0.9, 0.22, and 7%, respectively. Sodium chloride, lactate, potassium, and calcium ions are generally mixed into compound Ringer’s lactate solution. There is no clear conclusion or suggestion on which type of saline has a better therapeutic effect or fewer side effects, although some studies have presented their conclusions. One study pointed out that hypertonic saline has a strong effect on symptom improvement, especially in patients with allergic rhinitis, but also presents with many side effects, mainly such as a burning sensation and nasal irritation [48]. In terms of antibacterial effect, although not conclusive, Woods et al. believe that hypotonic saline and hypertonic saline are more effective than isotonic saline [49]. In terms of the postoperative quality-of-life score, Ringer’s lactate solution provided better results than other saline solutions [50].

Nasal decongestants are the most effective drugs to reduce nasal congestion. Although they are effective with a fast onset of action, adverse reactions are common [51]. Decongestants can constrict the nasal blood vessels, which subsequently reduce liquid extravasation, thereby reducing edema, mucus production, and nasal congestion [52]. The different nasally administered decongestants currently available in the market include oxymetazoline (Afrin^®^), xylometazoline (Otrivin^®^), and naphazoline (Privine^®^). Their excipients are essentially the same. An innovative study involving 108 patients confirmed the efficacy of imidazoline derivatives on nasal respiratory function. After the patients were divided into groups, they were given six different imidazole derivatives (oxymetazoline, xylometazoline, indanazoline, naphazoline, tramazoline, and tetryzoline). The maximum decongestant effect of all substances can be observed after 20–40 min, but the decongestant effect of tramazoline, xylometazoline, and oxymetazoline is more lasting, and seen within 4 h after administration, while oxymetazoline still has a significant decongestant effect after 8 h [53].

To date, the only vaccine that has been administered via the nose is the live attenuated influenza vaccine (LAIV). FluMist Quadrivalent^®^ is approved for use in the United States, and Fluenz Tetra^®^ is approved for use in Europe. There is little difference in their composition, and they have proven to be a safe and effective influenza vaccine [54]. This vaccine design mimics natural infection and provides mucosal immunity; therefore, it can be used as a large-scale vaccine [55].

### 2.2. Devices for Nasal Drug Delivery

Devices such as nose drops, nasal sprays, and nasal douches can be seen almost everywhere in the world. These devices are simple to use, and people are still tirelessly developing and improving these devices to achieve improved drug delivery.

Nasal drops are administered by drawing liquid into a glass dropper, inserting the dropper into the nostril, and squeezing the top rubber valve to release the liquid [56]. This inexpensive device, which does not require preservatives and improves the deposition of drugs in the nasal tract, is common in many products, such as in decongestants and normal saline [57,58]. However, this device is not popular because it requires a specific body posture, with the head tilted back and neck extended. For example, patients with CRS often have headaches and discomfort when lowering their heads, leading to poor patient compliance [59]. Therefore, for various reasons, nasal drops are used less frequently, and nasal sprays have become one of the most commonly used options for the treatment of nasal diseases.

Unlike nasal drops, nasal sprays provide measured doses containing active components that are dissolved or suspended in the excipient solution or in a mixture of non-pressurized dispensers for drug delivery to the nasal cavity. This device has the advantages of being non-invasive, avoiding the first-pass hepatic effect, with a fast onset of action, and good patient compliance [60]. For nasal spray products, it is important to include key parameters, such as single actuation content, droplet size distribution, and spray mode. These properties may affect drug distribution, including the deposit location, deposit surface area, and residence time of the drug in the nose, thereby affecting the absorption of the drug at its active site and the eventual systemic circulation. In a study on the effect of nozzle direction on spray droplet distribution in the nasal cavity, researchers found that the spray efficiency in the middle direction was higher than that in the upper or lower direction, and 10 μM was the most suitable particle size, because most agents that are formulated as a spray could be delivered to the target area [61]. In another study, a new nasal spray strategy was developed by reorienting the spray axis to make use of the inertial movement of particles and using a digital model based on medical imaging. When compared with the traditional spray technology, this strategy greatly improves the efficiency of drug delivery [62].

In addition to the abovementioned commonly used devices, some less common devices are available, which have their own advantages. Sonic nebulization devices can optimize aerosol deposits in the nasal cavity and effectively target the anatomical area of interest [63]. Mucosal atomization devices can convert liquid medicine into a fine mist and effectively transport it to deeper parts of the nasal cavity [64]. The sinus implant exerts an extended anti-inflammatory action by slowly diffusing corticosteroids to the surrounding mucosa [65].

### 2.3. Methods for Nasal Drug Delivery

The mucus layer of the nasal cavity is a hydrophilic absorption barrier. In nasal administration, the mucus adhesion system transports drug molecules to various mucous membranes by prolonging the residence time of drugs at the absorption site, and the mucus penetration system can realize a wider particle distribution and the deeper penetration of drug molecules [66]. Among the various nasal administration methods described below, some belong to mucus adhesion systems and some belong to mucus permeation systems.

Hydrogels are similar to natural tissue microenvironments because of their porous and hydrated molecular networks [67]. Hydrogels are widely used as matrix systems to control the release of macromolecules and can be combined with nanoparticles to design new systems, significantly improving the efficiency of drug absorption [68]. Although they have many advantages as a drug delivery system through the nasal administration route, their toxicity is a concern. Furthermore, their drug transport capacity in the nasal cavity or central nervous system has not been widely evaluated and needs to be studied in more detail in the future. Hyaluronic acid shows mucosal adhesion due to its high hydration capacity, and hyaluronic acid coatings can also enhance the permeability of the nasal mucosa. Therefore, hyaluronic acid-coated micelles can be considered for nasal drug delivery [69]. Some researchers have prepared self-assembled stable micelles with a polyion stable core, which consist of a mixture of methoxy PEG–PDLLA–polyglutamate and methoxy PEG–PDLLA–poly(l-lysine). It has the advantages of controlling drug release and improving carrier stability [70]. Other researchers have investigated the co-assembly of positively charged patchy micelles and negatively charged bovine serum albumin molecules. Patchy micelles are prepared using block copolymer brushes as templates, leading to the co-assembly of protein molecules into vesicular structures [71]. Lipid-based nanoparticles have many advantages as drug delivery systems, including their simple formulation and high bioavailability. Therefore, lipid-based nanoparticles are the most common nano-drug carriers to be approved by the FDA [72]. Liposomes can also bypass the blood–brain barrier to deliver drugs directly to the brain. The lipid solubility of drugs encapsulated in liposomes improves significantly, resulting in a higher bioavailability and efficiency [73]. Extensive research on liposomes will open up new opportunities for the nasal delivery of drugs and vaccines. Lipid-based nanoparticles are different from traditional liposomes mainly because they form a micellar structure in the particle core, which can be changed as needed [74]. Mohamed et al. [75] used new lipid-based nanoparticles incorporated into thermosensitive in situ gel for the intranasal administration of terbutaline sulfate and found that the permeability was three times that of the control group.

Nano suspension is a drug delivery method to enhance the solubility of drugs in the nasal mucosa [76]. Several studies have reported the nasal delivery of nanosuspensions. The researchers added carvedilol nanosuspensions to the in situ gel and found that they showed enhanced mucosal adhesion [77]. Other researchers prepared resveratrol-based nanosuspensions for brain delivery and achieved high-solubility drug delivery in nasal drug delivery [78]. The main components of nanoemulsions are oil, surfactants, cosurfactants, and the aqueous phases. If oil is used as an absorption enhancer, it can promote drug penetration through the nasal mucosa [79]. One study showed that specific polar lipids enhanced the penetration of drugs through the nasal mucosa, which may contribute to transcellular and paracellular pathways [80]. Many studies have shown that nasal nanoemulsions are an effective, non-invasive, and safe drug delivery system for direct nose-to-brain targeted drug delivery. Mahajan et al. used nanoemulsions to deliver anti-retroviral drugs to the brain via nasal administration. However, owing to its low solubility in water, its bioavailability is poor [81]. Yadav et al. evaluated nanoemulsions for encapsulating anti-tumor necrosis factor alpha siRNA, showing that it can achieve increased siRNA uptake in the brain [82]. These characteristics of nanoemulsions make them suitable for nose-to-brain delivery and represent a promising nose-to-brain drug delivery strategy. However, clinical studies on these preparations are still needed to prove their applicability in clinical practice [83].

The powder preparation of drugs has a better physicochemical stability than the solution preparation, and the concentration is higher after deposition onto the nasal mucosa. Nasal dry powder preparations have been shown to increase the residence time in the nasal cavity, thereby increasing the absorption of drugs by the mucosa [84]. Some researchers used the freeze-drying technology of polymer–drug solutions in order to prepare budesonide, an insoluble drug, as a dry powder. It was found that, compared with a water-based suspension, the release rate of freeze-dried preparation in nasal mucosa was faster and more suitable for the aerodynamic characteristics of nasal administration [85]. Microparticles, also known as micronized powders, range in size from 1–1000 μm. Most of the microparticles are produced by spray drying or spray freeze-drying. Most of their components are made of soluble excipients, which can be quickly dissolved in the nasal mucus and take effect quickly. In addition, the microparticles can be made of polymers that are encapsulated in the matrix structure of drug active substances, so as to maintain the drug release for a long time [86]. Inorganic nanoparticles with tunable and diverse characteristics have great potential in the field of nanomedicine. Inorganic nanoparticles can be designed with various sizes, structures, and geometric shapes after accurate preparation [87]. Owing to the characteristics of their basic materials, inorganic nanoparticles have unique physical, electrical, magnetic, and optical properties. For example, gold nanoparticles have free electrons on their surface, which oscillate continuously at a frequency, depending on their size and shape, giving them photothermal properties [88]. Microspheres are widely used in nasal drug delivery. Microspheres that are prepared by spray drying or emulsion crosslinking with cyclodextrins or chitosan as solubilizers and absorption enhancers significantly improved the in vivo bioavailability of encapsulated drugs, which enhanced drug permeation through respiratory and olfactory epithelium through transcellular transports or paracellular transport through olfactory epithelium cells [89,90]. It can prolong the contact time at the site of drug absorption because the surface of the microspheres has wrinkles and can adhere to the nasal mucosal epithelial cells [91]; it is also easily degraded by enzymes, and provides continuous drug release. Quantum dots are nanocrystals composed of semiconductor components that are very small in size and have charges on their surfaces, which makes them more permeable through tight connections [92]. A recent study found that the accumulation of carboxylate-modified quantum dots in the epithelial and submucosal regions of olfactory tissue increased by approximately 2.5-fold when compared with that in the respiratory tissue, showing its potential as a quantum dot carrier for nasal drug delivery [93]. Carbon quantum dots have a lower toxicity and better biocompatibility than gold and other semiconductor quantum dots [94]. A carbon quantum dot from sodium alginate developed by researchers for gene delivery applications had good plasmid DNA aggregation ability [95]. Another graphene quantum dot has been shown to enhance the intracellular absorption of formulated drugs [96]. Although several characteristics of carbon nanotubes make their application in the field of medicine very attractive, especially in drug delivery, some problems must be solved in order to apply them to clinical trials, such as their inherent cytotoxicity [97]. One study found that carbon nanotubes had harmful effects on normal human nasal epithelial cells after 12 days of exposure, and the differentiation function and oxidative stress of exposed nasal cells were adversely affected when compared to unexposed nasal cells [98].

Through various examples, it can be seen that both the mucus adhesion system and mucus penetration system show the potential of mucosal administration. The choice of which system depends on various aspects, including solubility, membrane permeability, mode of action, and the consideration of rapid or continuous release characteristics. Based on the above complex factors, it is not comprehensive to decide which system to adopt alone. Therefore, it is estimated that the future trend will be to combine the two systems into one, that is, a drug delivery system that has the characteristics of mucosal adhesion and mucosal penetration at the same time.

## 3. Vaccines for Nasal Administration

In addition to LAIV, which has been approved by the Food and Drug Administration (FDA) for more than 10 years, other nasal vaccines are also under research. A novel immunomodulatory drug developed for nasal administration [99] showed sustained antiviral and liver-protective properties in phase III clinical trials in patients with chronic hepatitis B. Vacc-4x is a therapeutic human immunodeficiency virus (HIV) vaccine based on GAg p24. A randomized controlled trial in 2014 proved that the intranasal administration of this formulation was safe [100]. In 2019, 33 participants from the United States and Europe were recruited and re-immunized, and the results showed that Vacc-4x was safe and well-tolerated, which attributes to HIV cure strategies [101]. Other intranasal vaccines are also being actively developed, such as AdVAV [102], an adenovirus vector vaccine that expresses protective antigens from *Bacillus anthracis*, a norovirus vaccine prepared in situ with dry gel powder [103], and a non-replicating vaccine against the respiratory syncytial virus [104].

### 3.1. Nasal Vaccines

In addition to the FDA-approved nasal vaccine, which is a liquid formulation, dry powder and gel-based vaccine formulations are receiving increasing attention in research (Figure 3) [105]. When compared with the shortcomings of liquid vaccines, which are susceptible to physical, chemical, and thermal instability [106], dry powder vaccines have the advantages of improved chemical and physical stability, which are more conducive to the preservation and transportation of the vaccine [107]. Different devices must be used to actively deliver the powder into the nasal cavity in order to effectively carry out the intranasal administration of a dry powder vaccine. At the time of writing, there is no FDA-approved dry powder vaccine for intranasal administration, but an increasing number of preclinical and clinical studies are underway, indicating that people are interested in this new method of vaccine formulation [108]. A large number of natural, synthetic, and semisynthetic gel-based nasal vaccines have also been developed. Gel-based nasal formulations have the advantage of increasing the retention rate of vaccines when compared with liquid formulations, and those that can adhere to the mucosal surface and specifically target memory or antigen-presenting cells are the most effective [109].

### 3.2. Methods of Nasal Vaccine Delivery

The research and development of intranasal vaccines are affected by various factors, such as inefficient antigen uptake, rapid clearance by nasal mucosal cilia, and difficulty in penetrating the epithelial barrier, which is caused by a large molecular size [110]. Therefore, it is particularly important to study different types of drug carriers that are suitable for nasal drug delivery systems [111], which can be divided into two categories: replicating and non-replicating delivery systems (Table 3).

In replicating delivery systems, when transgenic live viruses act as carriers, they can proliferate in the host tissue. After nasal administration, they have a positive effect in stimulating the immune response, which is typical of vesicular stomatitis virus [112], poliovirus [113], and influenza virus [114]. When living bacteria are designed as antigen carriers to stimulate the host immune system, they can stimulate lasting humoral and cellular immunity, trigger an innate immune response, and trigger an adaptive immune response [115]. Lactic acid bacteria (LAB) are relatively safe as mucosal delivery carriers because they are non-pathogenic, simple, and inexpensive. Several studies have demonstrated the feasibility of *Lactobacillus* as a carrier for delivering antigens directly to the nasal mucosa [116,117]. *Salmonella* is a pathogenic, Gram-negative, intracellular bacterium, but attenuated *Salmonella* has been widely used to transmit heterologous antigens and induce an immune response [118]. *Listeria* is a Gram-positive pathogen. *Listeria*, a human foodborne pathogen, has led to a large number of foodborne deaths caused by microorganisms [119]. One study showed that vaccines using attenuated *Listeria* as vectors are safe [120]. Although the virulence of attenuated strains is significantly reduced when compared to that of laboratory-created strains, safety problems still exist.

In a non-replicating delivery system, the carrier mimics the antigen of the immune system and causes similar uptake by antigen-presenting cells (APCs). The application of nanotechnology in mucosal vaccine delivery is an area of interest for many people, and in recent years, great achievements have been made and many original obstacles have been overcome [121]. The use of nanoparticles in vaccine preparations can improve antigen stability and immunogenicity, target delivery, the ability to cross the mucosal barrier, enhance the uptake of APCs, and prolong the availability of interaction with APCs [122]. Nanoparticle vaccines with many different components have been approved for use in humans, and their number has increased significantly over time.

The use of polymers as mucosal vaccine carrier systems is developing rapidly because they provide the advantage of delivering antigens to specific target sites, and they can control the release of antigens from the grasp of mucosal sites [123]. Therefore, polymers are usually studied for vaccine delivery through the nasal or oral cavity [124]. Although not successfully used in the clinical application of vaccines, polylactic-co-glycolic acid (PLGA) and polylactic acid (PLA) are the safest, most biodegradable, and most biocompatible polyester polymers currently available [125]. Researchers have demonstrated that the in vivo nasal residence time of PLGA nanoparticles was similar to that of soluble ovalbumin (OVA) protein [126]. Chitosan is a linear polysaccharide produced by the deacetylation of chitin [127]. Chitosan has many advantages, including its favorable biodegradability, high safety, and low toxicity [128]. It has been reported that the influenza hemagglutinin antigen that is presented through the nasal cavity of mice using chitosan as a carrier causes a significant response in the spleen of mice and has a 100% protection rate against influenza virus attack [129]. Lopes et al. showed that inactivated bronchitis viral antigen presented through the mouse nasal cavity using chitosan as a carrier caused significant specific IgA and IgG reactions in mice [130]. Dextran is a biocompatible, non-toxic, and non-immunogenic substance that is widely used in drug delivery and is often used as a mucosal absorption enhancer in drug delivery [131]. Tabassi et al. showed that dextran-carrying antigens could induce an effective immune response in rabbits following nasal delivery [132].

Liposomes are vesicles composed of one or more phospholipid membranes around an aqueous core [133]. In recent years, many studies have been devoted to developing liposomes as carriers for vaccine delivery systems [134,135]. An animal experiment reported that liposomes could be used for the nasal administration of an influenza vaccine. They successfully associated an antigen that had poor immunogenicity with liposomes and were able to induce a significantly higher specific immune response to the antigen [136]. Liposomes have low mucosal adhesion, which has been solved using cationic liposomes or by adjusting the surface properties of the liposomes with cationic polymers. Recently developed cationic liposomes have accelerated cellular internalization, resulting from a high membrane fusion ability. Cationic liposomes are composed of a classical lipid mixture of neutral lipid 1,2-dioleoyl-sn-glycerol-3-phosphoethanolamine (DOPE) and cationic lipid 1,2-dioleoyl-3-trimethylamine propane (DOTAP), which are used as carrier particles to achieve optimal cargo delivery through the attractive interactions between positively charged liposome surface and negatively charged cargo (such as protein, peptide, or RNA molecules) [137]. An animal experiment reported that the mRNA encoding a tumor-associated antigen encapsulated by a cationic liposome complex was released at the target site after its intranasal administration in mice, which induced an immune response and prevented tumor growth [138]. Polymer surface modification of liposomes can provide enhanced stability and the ability to stay in the nasal mucosa for a long time, in order to enhance the interaction between particles and immune cells, and these liposomes have a better stability [139]. When liposome–polymer hybrid nanoparticles are used to deliver antigens, they can improve vaccine efficacy, reduce dosage, and have a simple and low-cost manufacturing process [140]. A study found that chemically modified mRNA delivered by lipid-based nanoparticles restored chloride secretion in cystic fibrosis, and its nasal application restored chloride secretion to conductive airway epithelial cells in mice [141]. Lipopeptides have considerable potential in vaccine development and may play an important role. The advantages of using lipopeptides include their ease of design and the potential to customize highly specific reactions, which can be achieved through precision sequence engineering and lipopeptide structures [142]. The influenza virus antigen delivered by lipopeptides successfully produced an immune response in mice following its nasal administration [143].

Others include nan-emulsions, immunostimulating complexes (ISCOMs), and hydrogels. The nano-emulsion is composed of two immiscible liquids. It can transfer the apparent antigen directly to the surface of the mucosa and is highly stable [144]. A novel oil-in-water emulsion adjuvant containing squalane developed by Zhang et al. induced an effective immune response against the inactivated swine influenza virus. It had an effective protective effect and showed low toxicity in mice [145]. ISCOMs are composed of antigen, phospholipid, cholesterol, and other ingredients [146]. A vaccine based on ISCOMs showed a stronger mucosal and cellular immunity after its intranasal immunization [147]. The hydrogel nanoscale system has the characteristics of mucosal adhesion, enabling it to extend residence time, thereby increasing the contact time with the nasal mucosa and enhancing drug absorption [68]. Bedford et al. found that intranasal immunization with a chitosan hydrogel vaccine prolonged the retention time of antigens in the nasal mucosa [148].

## 4. Future Prospects

An increasing number of non-clinical and clinical studies support the feasibility and safety of drugs and vaccines administered via the nasal route. This alternative method of drug administration may address many unsolved medical problems and may lead to simpler and more efficient medical solutions. Innovations in different equipment and formulations will bring different properties to different nasal drug delivery products. Therefore, it is necessary to conduct in-depth research on drug deposition, experimental models, simulated use, and various single or composite materials as transmitters. Especially in the optimization of treatment delivery, nanoparticles are being studied as a current hotspot in order to overcome the problems of biological barriers. They provide the advantage of delivering antigens to specific targets and can control the release of antigens from mucosal sites. Therefore, they can be selectively modified and optimized in order to adapt to specific user requirements and scenarios to achieve better drug delivery. Despite extensive research in this field, nanoparticle-based products for intranasal delivery are not yet available. All these problems require cooperative research worldwide to solve them effectively.

## Figures and Tables

**Figure 1 pharmaceutics-14-01073-f001:**
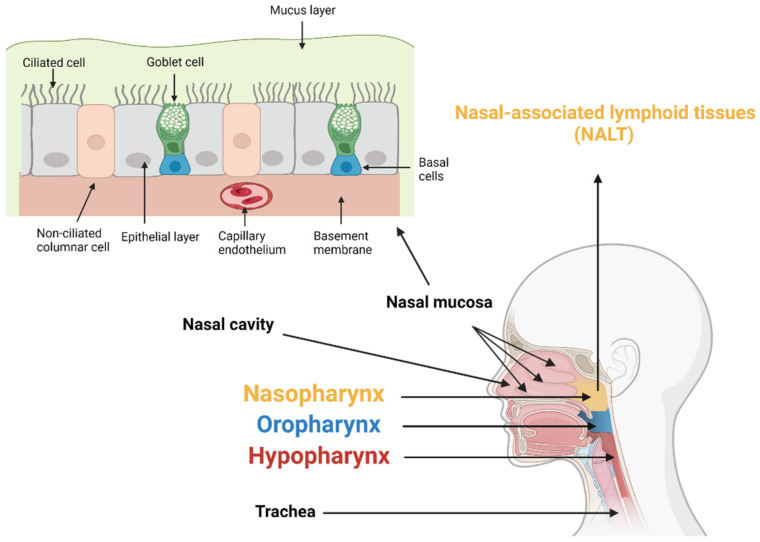
Schematic diagram of the nose, including the main cell types, and characteristic absorption barriers.

**Figure 2 pharmaceutics-14-01073-f002:**
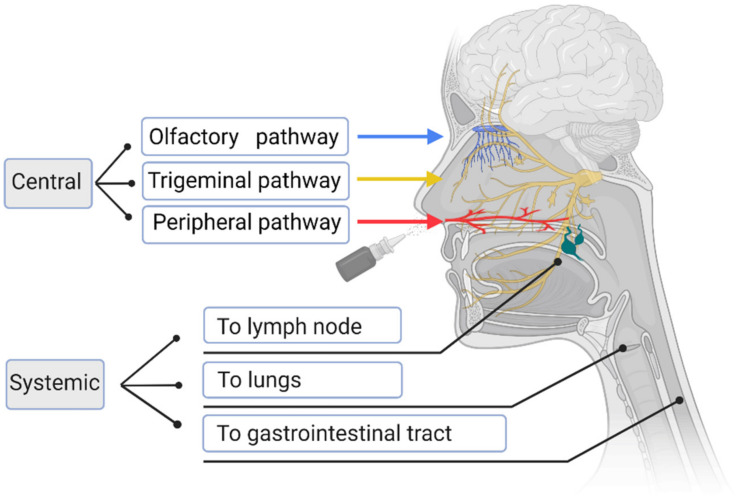
Nasal drug pathways and destinations.

**Figure 3 pharmaceutics-14-01073-f003:**
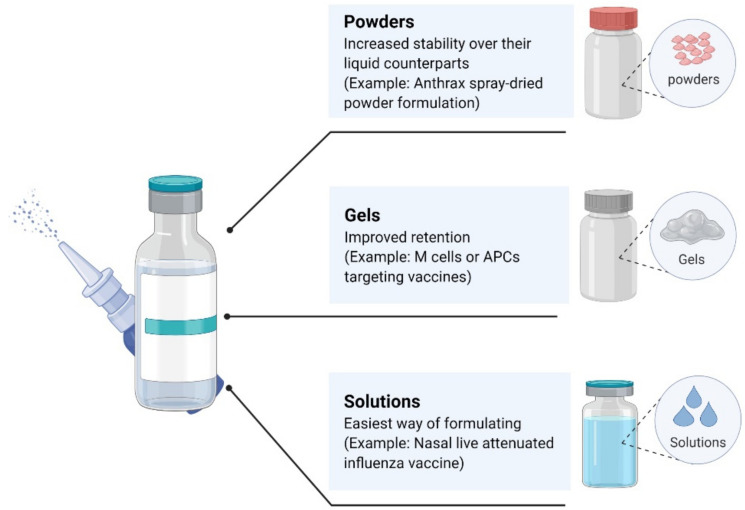
Different types of vaccine formulations for nasal delivery.

**Table 1 pharmaceutics-14-01073-t001:** Advantages and disadvantages of nasal drug delivery.

Advantages	Disadvantages
Rapid onset of action	Drug elimination
Less drug degradation	Low bioavailability
High rate of absorption	Irreversible damage of nasal mucosa
High patient compliance	Drug dose loss due to improper use
Self-administration by patients	The state of the nasal cavity affects the absorption of drugs
Direct nose-to-brain delivery	Unclear mechanism
Non-invasive drug delivery	Limited dose

**Table 2 pharmaceutics-14-01073-t002:** Commonly administered drugs via the nasal route of delivery.

Compounds	Types	Brand and Formulation
Corticosteroids	Ciclesonide	Omnaris^®^ (ciclesonide, hypromellose, potassium sorbate, and edetate sodium)
	Mometasone furoate	Nasonex^®^ (mometasone furoate, glycerin, sodium citrate, citric acid, and polysorbate 80)
	Fluticasone furoate	Avamys^®^ (fluticasone furoate, dispersible cellulose, polysorbate 80, benzalkonium chloride, and disodium edetate)
	Fluticasone propionate	Flonase^®^ (fluticasone propionate, microcrystalline cellulose, carboxymethylcellulose sodium, dextrose, and polysorbate 80)
Saline	Isotonic	Hospira Inc., Lake Forest, CA, USA (0.9% NaCl solutions)
	Hypotonic	Baxter Inc., Deerfield, IL, USA (0.22% NaCl solutions)
	Hypertonic	Nephron Inc., West Columbia, SC, USA (7% NaCl solutions)
	Ringer’s lactate solution	130, 109, 28, 4 and 3 mEq of sodium, chloride, lactate, potassium, and calcium ion in one liter of Ringer’s lactate solution.
Decongestants	Oxymetazoline	Afrin^®^ (oxymetazoline, povidone, edetate disodium, propylene glycol, and polyethylene glycol)
	Xylometazoline	Otrivin^®^ (xylometazoline, disodium edetate, sodium chloride, sorbitol, and benzalkonium chloride)
	Naphazoline	Privine^®^ (naphazoline, monobasic sodium phosphate, benzalkonium chloride, and disodium edetate)
Antihistamines	Azelastine	Astelin^®^ (azelastine, benzalkonium chloride, hypromellose, citric acid, and edetate disodium)
		Astepro^®^ (azelastine, benzalkonium chloride, hypromellose, sorbitol, and edetate disodium)
Vaccines	Live attenuated influenza virus	FluMist Quadrivalent^®^ (USA) (Live attenuated influenza virus reassortants, sucrose, gelatin, and dibasic potassium phosphate)
		Fluenz Tetra^®^ (Europe) (Live attenuated influenza virus reassortants, sucrose, gelatin, and dipotassium phosphate)

**Table 3 pharmaceutics-14-01073-t003:** Various carriers of different systems for nasal vaccine delivery.

Delivery System		Carrier		
Replicating delivery system	Virus	Vesicular stomatitis virus	Poliovirus	Influenza virus
	Bacteria	*Lactobacillus*	*Salmonella*	*Listeria*
Non-replicating delivery system	Polymer	Polyesters	Chitosan	Dextran
	Liposome	Cationic liposomes	Liposome–polymer hybrid nanoparticles	Lipopeptides
	Others	Nano-emulsion	ISCOMs	Hydrogel

## Data Availability

Not applicable.

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
