# Peer review of "Different Methods and Formulations of Drugs and Vaccines for Nasal Administration"

_pharmaceutics, 2022, doi:10.3390/pharmaceutics14051073_

Round 1

Reviewer 1 Report

General comments:

This is a very general and pharmaceutics text-book like review, that might be useful for students and new-comers to the field. Although the review is generally very well written, there are some unclear sentences in the text that requires the authors attention. In the abstract, the authors write that they will discuss nasal anatomy and mucosal environment, but very little is in fact presented in the text. For example for vaccines, it is essential to describe in more details the nasal-associated lymphoid tissue (NALT). In general, more critical reflections and discussions are needed. For example, what are the limitations of nasal drug delivery? The authors should also describe clearly the difference between mucoadhesive versus mucus-penetrating systems in the context of nasal delivery, and the critical discussion of these systems should be supported by examples.

Specific comments:

Abstract:

Safety is a key issue in the field of nasal drug delivery, and safety should be mentioned already in the abstract of the review. The abstract is very short and general. Additional points to include in the abstract are pros and cons of nasal drug delivery, as well as a critical discussion of nasal drug delivery technologies.

  1. Introduction:

Line 20: It is not clear what the authors mean by “rational” method of topical administration. What is an irrational method?

Line 21: Nasal administration is not characterized by a large surface area, as compared to for example pulmonary administration. So large, as compared to what?

Line 24: “less drug degradation”, as compared to what?

Line 35: Ii is too strong to say that “mucociliary clearance has been solved”. Please rephrase the sentence.

Lines 37/38: Avoid mentioning specific author names in the text.

Line 38: Usually, the bioavailability of the carrier is not measured, but the bioavailability of the drug. What was the drug in this case, and how was the bioavailability measured?

Line 39: What is “a suitable release curve”?

Line 52: Vaccines do not necessariy have to enter the systemic circulation. They can also activate a local, mucosal immue responses.

Lines 56/57: The sentence “The olfactory nerve can quickly deliver drugs…” is wrong. Drugs can be delivered to the olfactory bulb via the olfactory nerve.

  1. Drugs:

Line 75: I would not call nasal saline a drug. Please modify the sentence “various concentrations of saline”

Line 77: “work quickly” should be “fast on-set of action”

Line 79-80: “efficient at low doses” should be “potent”

Line 84: “it” should be “they”

Lines 125/126: A device cannot deliver a drug by itself, but it can be used to deliver a drug.

Lines 161-165: These sentences do not really tell the reader anything because they are very general.

  1. Vaccines

Lines 232/233: This sentences does not really tell the reader anything because it is very general.

Line 272: Emulsions and ISCOMs are not composed of polymers but of lipids.

Line 234-235: It is wrong to refer to polymers as the most mature carriers. Lipids are more commonly used

Lines 235-236: What is meant specifically by “appropriate size”? Please include references for this statement

Line 258: Which cationic liposomes (mention specific composition + physicochemical properties)

Line 260: “Stronger anti-degradation” is “enhance stability” or something like that

  1. ?? Where is section 4?

  1. Future prospects:

This section should present the future prospects of the field. Therefore, all the new things introduced in this section should be me moved to the previous sections.

Lines 285-286: Are the authors here referring to local acting drugs, or drugs with a systemic effect?

Lines 287-290: This sentence is not clear. Precise control? Control of what? Plasticity in size? Simple formulation? In general, nanomedicines are complex formulations. Higher bioavailability than what? Avoid relative and non-scientific words. This section is not presenting future prospects.

Line 296: It is too strong to call polymeric nanoparticles “ideal candidates”. Nothing is ideal. All systems have pros and cons, and they should be discussed.

Lines 309-310: The meaning of this sentence is unclear. Please improve.

Lines 335-345: The section about carbon nanotubes can be shortened. Very few people work with carbon nanotubes anymore.

Line 347: Please use the word for “spherical platform”, i.e. liposomes.

Line 354: What is meant by “good antibody response”? Avoid relative and non-scientific words.

  1. Conclusions

This section should be improved to reflect the points mentioned above.

Author Response

Response to Reviewer 1 Comments

We thank the reviewers for their constructive suggestions, which helped us revise the manuscript. A point-by-point response is attached below.

Black: reviewer’s comment

Red: author’s response

Blue: text in the revised manuscript

Point 1: This is a very general pharmaceutics text-book like review, that might be useful for students and newcomers to the field. Although the review is generally very well written, there are some unclear sentences in the text that requires the author’s attention. In the abstract, the authors write that they will discuss nasal anatomy and mucosal environment, but very little is in fact presented in the text. For example for vaccines, it is essential to describe in more detail the nasal-associated lymphoid tissue (NALT). In general, more critical reflections and discussions are needed. For example, what are the limitations of nasal drug delivery? The authors should also describe clearly the difference between mucoadhesive versus mucus-penetrating systems in the context of nasal delivery, and the critical discussion of these systems should be supported by examples.

Response 1: Thank you for your comments. The contents related to nasal anatomy and mucosal environment are supplemented in section 1.1. Mucosal environment of the nasal cavity (line 75), which also includes the introduction of NALT. The limitations of nasal administration are supplemented in Table 1 (line 60). The description of the mucoadhesive versus mucus-penetrating systems is reflected in section 2.3. Methods for nasal drug delivery (line 291).

Point 2: Abstract: Safety is a key issue in the field of nasal drug delivery, and safety should be mentioned already in the abstract of the review. The abstract is very short and general. Additional points to include in the abstract are pros and cons of nasal drug delivery, as well as a critical discussion of nasal drug delivery technologies.

Response 2: Thank you for your comments. The abstract was modified to mention safety and some other advantages and disadvantages of nasal administration (line 9).

Point 3: Line 20: It is not clear what the authors mean by “rational” method of topical administration. What is an irrational method?

Response 3: Thank you for your comments. The corresponding sentences have been improved.

Line 23: The nose is a very valuable route of administration, and the high vascularization and high permeability of nasal mucosa also make it possible to administer drugs through this route.

Point 4: Line 21: Nasal administration is not characterized by a large surface area, as compared to for example pulmonary administration. So large, as compared to what?

Response 4: Sorry for the unclear information. The sentence has been rewritten.

Line 28: Nasal administration has many advantages compared with oral administration, such as a rapid onset of action, less drug degradation, high rate of absorption. And compared with intravenous administration, it has high patient compliance, self-administration by patients, and direct nose-to-brain delivery by bypassing the blood-brain barrier via the olfactory nerve pathways.

Point 5: Line 24: “less drug degradation”, as compared to what?

Response 5: Sorry for the unclear information. The sentence has been rewritten. It is compared with oral administration.

Line 28: Nasal administration has many advantages compared with oral administration, such as a rapid onset of action, less drug degradation, high rate of absorption. And compared with intravenous administration, it has high patient compliance, self-administration by patients, and direct nose-to-brain delivery by bypassing the blood-brain barrier via the olfactory nerve pathways.

Point 6: Line 35: Ii is too strong to say that “mucociliary clearance has been solved”. Please rephrase the sentence.

Response 6: Thank you for your advice. The sentence has been rephrased.

Line 43: The use of mucosal adhesive overcomes some obstacles to mucociliary clearance.

Point 7: Lines 37/38: Avoid mentioning specific author names in the text.

Response 7: Thank you for your correction. The authors’ names are replaced by words such as researchers (line 47/50).

Point 8: Line 38: Usually, the bioavailability of the carrier is not measured, but the bioavailability of the drug. What was the drug in this case, and how was the bioavailability measured?

Response 8: Sorry for the confusing description, the sentence has been modified.

Line 47: Some researchers have prepared polymer-coated liposomes with mucosal adhesion, which can improve the bioavailability of drugs.

Point 9: Line 39: What is “a suitable release curve”?

Response 9: Sorry for the vague description, the sentence has been improved.

Line 51: slow release of encapsulated antigen.

Point 10: Line 52: Vaccines do not necessarily have to enter the systemic circulation. They can also activate a local, mucosal immune responses.

Response 10: Thank you for your reminder. The sentence has been improved.

Line 70: and vaccines that can act by activating local mucosal immune responses.

Point 11: Lines 56/57: The sentence “The olfactory nerve can quickly deliver drugs…” is wrong. Drugs can be delivered to the olfactory bulb via the olfactory nerve.

Response 11: Thank you for your correction. The error has been corrected.

Line 102: they can be transported to the olfactory bulb through the olfactory nerve.

Point 12: Line 75: I would not call nasal saline a drug. Please modify the sentence “various concentrations of saline”

Response 12: Thank you for your reminder. The sentence has been modified

Line 143: Many drugs are administered through the nasal route, including corticosteroids, decongestants, antihistamines, and vaccines. In addition, various concentrations of saline are also used through the nasal route.

Point 13: Line 77: “work quickly” should be “fast on-set of action”

Response 13: Thank you for your suggestion. The sentence has been revised.

Line 149: They are fast on-set of action

Point 14: Line 79-80: “efficient at low doses” should be “potent”

Response 14: Thank you for your correction. That word has been revised.

Line 149: they are potent at low doses

Point 15: Line 84: “it” should be “they”

Response 15: Thank you for your reminder. The error has been corrected.

Line 162: they exert an anti-inflammatory effect through trans-activation or trans-inhibition.

Point 16: Lines 125/126: A device cannot deliver a drug by itself, but it can be used to deliver a drug.

Response 16: Thank you for your reminder. The sentence has been corrected.

Line 249: These devices are simple to use but cannot be effectively used to deliver drugs

Point 17: Lines 161-165: These sentences do not really tell the reader anything because they are very general.

Response 17: Thank you for your correction. After consideration, this paragraph has been deleted

Point 18: Lines 232/233: This sentence does not really tell the reader anything because it is very general.

Response 18: Thank you for your reminder. The sentence has been revised.

Line 442: The use of polymers as mucosal vaccine carrier systems is developing rapidly because they provide the advantage of delivering antigens to specific target sites, and they can control the release of antigens from the grasp of mucosal sites.

Point 19: Line 272: Emulsions and ISCOMs are not composed of polymers but of lipids.

Response 19: Thank you for pointing out. I wanted to write the word "other" instead of other polymers. The sentence has been modified.

Line 487: Others include nano-emulsions, immunostimulating complexes (ISCOMs), and hydrogels.

Point 20: Line 234-235: It is wrong to refer to polymers as the most mature carriers. Lipids are more commonly used

Response 20: Thank you for your correction. The wrong sentence has been deleted.

Point 21: Lines 235-236: What is meant specifically by “appropriate size”? Please include references for this statement

Response 21: Thank you for your reminder. The whole sentence has been revised.

Line 442: The use of polymers as mucosal vaccine carrier systems is developing rapidly because they provide the advantage of delivering antigens to specific target sites, and they can control the release of antigens from the grasp of mucosal sites.

Point 22: Line 258: Which cationic liposomes (mention specific composition + physicochemical properties)

Response 22: Thank you for your reminder. The description has been reflected in the article.

Line 471: Cationic liposomes are composed of a classical lipid mixture of neutral lipid 1,2-dioleoyl-sn-glycerol-3-phosphoethanolamine (DOPE) and cationic lipid 1,2-dioleoyl-3-trimethylamine propane (DOTAP) are used as carrier particles to achieve optimal cargo delivery through attractive interaction between positively charged liposome surface and negatively charged cargo (such as protein, peptide or RNA molecules).

Point 23: Line 260: “Stronger anti-degradation” is “enhance stability” or something like that

Response 23: Thank you for your reminder. Changes have been made.

Line 476: Polymer surface modification of liposomes can provide enhanced stability

Point 24: Where is section 4?

Response 24: I'm very sorry, it's a mistake in my writing. There is no section 4. The original section 5 should have been section 4, and all errors have been corrected.

Point 25: This section should present the future prospects of the field. Therefore, all the new things introduced in this section should be moved to the previous sections.

Response 25: Yes, there are some problems with my typesetting. So part of this section was moved to section 2.3. Methods for nasal drug delivery (line 291).

Point 26: Lines 285-286: Are the authors here referring to local acting drugs, or drugs with a systemic effect?.

Response 26: It originally referred to drugs with systemic effects, but this paragraph was deleted when the section was rearranged.

Point 27: Lines 287-290: This sentence is not clear. Precise control? Control of what? Plasticity in size? Simple formulation? In general, nanomedicines are complex formulations. Higher bioavailability than what? Avoid relative and non-scientific words. This section is not presenting future prospects.

Response 27: Thank you for your correction. This paragraph was deleted when the section was rearranged

Point 28: Line 296: It is too strong to call polymeric nanoparticles “ideal candidates”. Nothing is ideal. All systems have pros and cons, and they should be discussed.

Response 28: Thank you for your suggestion. The description has changed

Line 511: Although polymers have the disadvantages of toxicity and aggregation, they are considered to be one of the choices for the preparation of macroscopic dimensions functional materials because of their unique physical and chemical properties, such as the change of polymer length can lead to the change of crystal morphology.

Point 29: Lines 309-310: The meaning of this sentence is unclear. Please improve.

Response 29: Thank you for your correction. A detailed description has been made.

Line 517: Microspheres prepared by spray drying or emulsion crosslinking with cyclodextrins or chitosan as solubilizers and absorption enhancers significantly improved the in vivo bioavailability of encapsulated drugs, which enhanced drug permeation through respiratory and olfactory epithelium through transcellular transports or paracellular transport through olfactory epithelium cells.

Point 30: Lines 335-345: The section about carbon nanotubes can be shortened. Very few people work with carbon nanotubes anymore.

Response 30: Thank you for your suggestion. The content has been reduced (Line 546)

Point 31: Line 347: Please use the word for “spherical platform”, i.e. liposomes.

Response 31: Thank you for your suggestion. The description has been modified.

Line 557: Lipid-based nanoparticles have various structures, but the most typical is the liposomes.

Point 32: Line 354: What is meant by “good antibody response”? Avoid relative and non-scientific words.

Response 32: I'm very sorry for the unscientific description, the description has been modified.

Line 562: They successfully associated an antigen with poor immunogenicity with liposomes and were able to induce a significantly higher specific immune response to the antigen

Point 33: This section should be improved to reflect the points mentioned above.

Response 33: Thank you for your suggestion. Some modifications have been made.

Line 583: They provide the advantage of delivering antigens to specific targets and can control the release of antigens from mucosal sites.

Sincerely,

Tae Hoon Kim, M.D., PhD.(corresponding author)

Professor of Otorhinolaryngology-Head and Neck Surgery

Korea University College of Medicine

Director, External Communication/Cooperation Team, Anam Hospital

Director, International Medical Device Clinical Trial Support Center,

Korea University Medicine 73, Goryedae-ro, Seongbuk-gu, Seoul 02841, Korea

73, Goryeodae-ro, Seongbuk-gu, Seoul 02841, Korea

TEL: (82)-2-920-6405, (82)-10-9491-9886
FAX: (82)-2-925-5233
E-mail: [email protected]

Reviewer 2 Report

The manuscript pharmaceutics-1679315 reviews the current knowledge related to the nasal route of drugs administration, and presents the nasal drugs and vaccines, as well as different materials and methods of drug delivery.

In my opinion, the paper is well organized, contain interesting results, but still need some improvements before to be published in the Pharmaceutics journal.

Abstract

- L. 11-12: "The current review introduces the nasal anatomy and mucosal environment"?  Maybe the authors want to say “The current review presents” or “brings into discussion”! Please revise this sentence!

- Moreover, where are presented within the manuscript the information related to “nasal anatomy and mucosal environment”? Please add a separate section where to present all this information together with the four characteristic absorbing barriers, so that the manuscript corresponds to the description in the abstract!

  1. Introduction

- L. 50-51: “This review describes the mucosal environment for nasal administration”!! The same observation to add a separate paragraph where the authors must mention the anatomy of the human nasal cavity, as well as the four principal cell types (goblet, ciliated, non-ciliated columnar and basal cells) and the four characteristic absorbing barriers (mucus layer, epithelial layer, interstitium and basement membrane, and capillary endothelium)! An explicit figure of the anatomy of the human nasal cavity is welcome!

- Please add more information about the limitations of nasal drug administration, such as the low bioavailability, due to the metabolism at mucosal surface, or the risk of local side effect, as the irreversible damage of nasal mucosa, due to the constituents added to the formulations and also, the fact that it is not applicable to all drugs or other different disadvantages.

- It would be interesting to make a table where to present the advantages and the disadvantages of this route for drug administration!

- Please emphasize the novelty of this manuscript compared to the other works which can be found in the literature!

1.1. Pathways and destinations of nasally administered drugs

- Please provide substantial information in this section! The authors cannot present each pathway by a single sentence! Each pathway must be in-depth explained and expanded accordingly! This section is of great importance for the readers to understand this review!

- Please add information about the route of each pathway shown in Figure 3 mentioning the olfactory neurons and supporting cells!

  1. Drugs for nasal administration

- L. 80: “fewer adverse effects compared with other drugs”! Please mention these adverse effects!

- There is nothing mentioned about neurodegenerative diseases and brain cancer! Please add information also about these diseases!

2.1. Formulations of nasal drugs

- L. 83-95: Antihistamines act by blocking the effects of histamine, that is responsible for many allergic symptoms, while corticosteroids belong to the glucocorticoid drug class and are mostly use for their strong anti-inflammatory effects. Please explain the mode of action for each class and make a comparation between them! Moreover, please explain the efficiency for each type of drug from Table 1, as a results of clinical studies. These results must be added within the manuscript!

- L. 106-107: “presents with many side effects”! Please mention these effects!

- L. 117-118: “To date, the only vaccine that has been administered”! This sentence must be a new paragraph, due to the fact that presents a new formulation.

2.2. Devices for nasal drug delivery

- Why the authors did not present a section related to the “Methods of nasal drug delivery” and did this only for the vaccines? The authors presented only the devices, nasal drops and nasal spray and briefly mentioned sonic nebulization devices and mucosal atomization devices!

However, there are different known methods which were not presented, such as nasal gels, nasal micellar formulations, nasal liposomal formulations, nasal suspensions, nasal emulsions, nasal powders, nasal microparticles, nasal nanoparticles. Please add all this information due to the fact that the title of the manuscript is “Different methods and formulations of drugs and vaccines for nasal administration” and the authors must respond to the proposed title!

Even if there are some information presented in the section “5. Future prospects”, they must also be presented in the section “Methods of nasal drug delivery” section, with benefits and ways of action!

Author Response

Response to Reviewer 2 Comments

We thank the reviewers for their constructive suggestions, which helped us revise the manuscript. A point-by-point response is attached below.

Black: reviewer’s comment

Red: author’s response

Blue: text in the revised manuscript

Point 1: L. 11-12: "The current review introduces the nasal anatomy and mucosal environment"?  Maybe the authors want to say “The current review presents” or “brings into the discussion”! Please revise this sentence!.

Response 1: Thank you for your correction. The error has been corrected.

Line 14: The current review presents the nasal anatomy and mucosal environment for nasal delivery of vaccines and drugs.

Point 2: Moreover, where is presented within the manuscript the information related to “nasal anatomy and mucosal environment”? Please add a separate section where to present all this information together with the four characteristic absorbing barriers, so that the manuscript corresponds to the description in the abstract!

Response 2: I'm very sorry for the lack of my content. A separate section 1.1. Mucosal environment of nasal cavity has been added (line 75)

Point 3: L. 50-51: “This review describes the mucosal environment for nasal administration”!! The same observation to add a separate paragraph where the authors must mention the anatomy of the human nasal cavity, as well as the four principal cell types (goblet, ciliated, non-ciliated columnar, and basal cells) and the four characteristic absorbing barriers (mucus layer, epithelial layer, interstitium and basement membrane, and capillary endothelium)! An explicit figure of the anatomy of the human nasal cavity is welcome!

Response 3: Thank you for your suggestion. A new figure has been added to describe the anatomical structure of the human nasal cavity, main cell types, and characteristic absorption barriers (Figure 1) (Line 94-97).

Point 4: Please add more information about the limitations of nasal drug administration, such as the low bioavailability, due to the metabolism at the mucosal surface, or the risk of local side effects, such as the irreversible damage of nasal mucosa, due to the constituents added to the formulations and also, the fact that it is not applicable to all drugs or other different disadvantages.

Response 4: Thank you for your suggestions. The corresponding contents have been supplemented.

Line 56: In addition to the above disadvantages, as shown in Table 1, nasal drug delivery also has the disadvantages like irreversible damage to nasal mucosa caused by the ingredients added in the formula, not applicable to all drugs, and affected by nasal conditions such as allergy conditions.

Point 5: It would be interesting to make a table where to present the advantages and the disadvantages of this route for drug administration!

Response 5: Your suggestion is very good. A table has been added (Table 1) (line 60).

Point 6: Please emphasize the novelty of this manuscript compared to the other works which can be found in the literature!

Response 6: Thank you for your suggestion. The corresponding content has been added.

Line 68: This review comprehensively describes the mucosal environment for nasal administration, explains and classifies the drugs entering the systemic circulation via the nasal route of delivery and vaccines that can act by activating local mucosal immune responses, and introduces newly developed and developing materials for nasal drug carriers. The advantages and disadvantages of nasal drug delivery methods are analyzed, which provided some information and tips for the study of new nasal administration methods.

Point 7: Please provide substantial information in this section! The authors cannot present each pathway in a single sentence! Each pathway must be in-depth explained and expanded accordingly! This section is of great importance for the readers to understand this review! Please add information about the route of each pathway shown in Figure 3 mentioning the olfactory neurons and supporting cells!

Response 7: Thank you for your correction. I quite agree with you. Relevant explanations and extensions have been added to the content (line 99-134).

Point 8: L. 80: “fewer adverse effects compared with other drugs”! Please mention these adverse effects!

Response 8: Thank you for your suggestion. The content has been reflected in the sentence.

Line 151: they have fewer adverse effects such as sedation, drowsiness, amnesia, and respiratory depression.

Point 9: There is nothing mentioned about neurodegenerative diseases and brain cancer! Please add information also about these diseases!

Response 9: Thank you for your suggestion. Relevant content has been added.

Line 154: Effective delivery of drugs to the central nervous system is the key to the treatment of brain tumors and neurodegenerative diseases such as stroke, Parkinson's disease, and Alzheimer's disease. Because intranasal delivery can make drugs pass through the blood-brain barrier, which is one of the most important barriers in the central nervous system, it can be considered as a development potential local delivery strategy.

Point 10: L. 83-95: Antihistamines act by blocking the effects of histamine, that is responsible for many allergic symptoms, while corticosteroids belong to the glucocorticoid drug class and are mostly use for their strong anti-inflammatory effects. Please explain the mode of action for each class and make a comparation between them! Moreover, please explain the efficiency for each type of drug from Table 1, as a results of clinical studies. These results must be added within the manuscript!

Response 10: Thank you for your suggestion. A lot of content have been added to describe the mode of action of various drugs and their efficacy in clinical research (line 161-246)

Point 11: L. 106-107: “presents with many side effects”! Please mention these effects!

Response 11: Thank you for your suggestion. The details have been reflected in the sentence.

Line 221: presents with many side effects mainly such as burning sensation and nasal irritation.

Point 12: L. 117-118: “To date, the only vaccine that has been administered”! This sentence must be a new paragraph, due to the fact that presents a new formulation.

Response 12: Thank you for your suggestion. Changes have been made (line 241).

Point 13: Why the authors did not present a section related to the “Methods of nasal drug delivery” and did this only for the vaccines? The authors presented only the devices, nasal drops and nasal spray and briefly mentioned sonic nebulization devices and mucosal atomization devices!

However, there are different known methods which were not presented, such as nasal gels, nasal micellar formulations, nasal liposomal formulations, nasal suspensions, nasal emulsions, nasal powders, nasal microparticles, nasal nanoparticles. Please add all this information due to the fact that the title of the manuscript is “Different methods and formulations of drugs and vaccines for nasal administration” and the authors must respond to the proposed title!

Even if there are some information presented in the section “5. Future prospects”, they must also be presented in the section “Methods of nasal drug delivery” section, with benefits and ways of action!

Response 13: Thank you for your advice. A new paragraph describing information related to nasal gels, nasal micellar formulations, nasal liposomal formulations, nasal suspensions, nasal emulsions, nasal powders, nasal microparticles, and nasal nanoparticles was added. (2.3. Methods for nasal drug delivery) (line 291-374).

Sincerely,

Tae Hoon Kim, M.D., PhD.(corresponding author)

Professor of Otorhinolaryngology-Head and Neck Surgery

Korea University College of Medicine

Director, External Communication/Cooperation Team, Anam Hospital

Director, International Medical Device Clinical Trial Support Center,

Korea University Medicine 73, Goryedae-ro, Seongbuk-gu, Seoul 02841, Korea

73, Goryeodae-ro, Seongbuk-gu, Seoul 02841, Korea

TEL: (82)-2-920-6405, (82)-10-9491-9886
FAX: (82)-2-925-5233
E-mail: [email protected]

Round 2

Reviewer 2 Report

In my opinion, the authors paid attention to all comments, have made improvements to the manuscript pharmaceutics-1679315, as compare to previous version, and I recommend the publication of the present version in Pharmaceutics journal.  

Author Response

Thank you again for your kind comments on our manuscript. The manuscript has made a great improvement after modification according to your suggestion.